# Chronic Intermittent Hypoxia-Induced Diaphragm Muscle Weakness Is NADPH Oxidase-2 Dependent

**DOI:** 10.3390/cells12141834

**Published:** 2023-07-12

**Authors:** Sarah E. Drummond, David P. Burns, Sarah El Maghrani, Oscar Ziegler, Vincent Healy, Ken D. O’Halloran

**Affiliations:** Department of Physiology, School of Medicine, College of Medicine and Health, University College Cork, T12 XF62 Cork, Ireland

**Keywords:** obstructive sleep apnoea, diaphragm, chronic intermittent hypoxia, NADPH oxidase

## Abstract

Chronic intermittent hypoxia (CIH)-induced redox alterations underlie diaphragm muscle dysfunction. We sought to establish if NADPH oxidase 2 (NOX2)-derived reactive oxygen species (ROS) underpin CIH-induced changes in diaphragm muscle, which manifest as impaired muscle performance. Adult male mice (C57BL/6J) were assigned to one of three groups: normoxic controls (sham); chronic intermittent hypoxia-exposed (CIH, 12 cycles/hour, 8 h/day for 14 days); and CIH + apocynin (NOX2 inhibitor, 2 mM) administered in the drinking water throughout exposure to CIH. In separate studies, we examined sham and CIH-exposed NOX2-null mice (B6.129S-*Cybb^TM1Din^*^/J^). Apocynin co-treatment or NOX2 deletion proved efficacious in entirely preventing diaphragm muscle dysfunction following exposure to CIH. Exposure to CIH had no effect on NOX2 expression. However, NOX4 mRNA expression was increased following exposure to CIH in wild-type and NOX2 null mice. There was no evidence of overt CIH-induced oxidative stress. A NOX2-dependent increase in genes related to muscle regeneration, antioxidant capacity, and autophagy and atrophy was evident following exposure to CIH. We suggest that NOX-dependent CIH-induced diaphragm muscle weakness has the potential to affect ventilatory and non-ventilatory performance of the respiratory system. Therapeutic strategies employing NOX2 blockade may function as an adjunct therapy to improve diaphragm muscle performance and reduce disease burden in diseases characterised by exposure to CIH, such as obstructive sleep apnoea.

## 1. Introduction

Obstructive sleep apnoea syndrome (OSAS) represents the most frequently diagnosed form of sleep disordered breathing (SDB) [1]. Repetitive upper airway collapse throughout the night cycle gives rise to periodic oxygen desaturations, with the frequency of events defining mild, moderate, and severe sleep apnoea [2,3]. Chronic intermittent hypoxia (CIH), arising as a consequence of repetitive airway collapse, is widely considered to underlie the deleterious morbidities reported in patients with OSAS. Diaphragm muscle dysfunction has been reported in OSAS patients [4,5]. A fine balance between the dilating and stabilizing forces produced by upper airway muscles and the sub-atmospheric collapsing pressure of the diaphragm ensures that the upper airway calibre remains patent [6]. Impaired diaphragm function may contribute to a reduced ability to generate large powerful manoeuvres necessary for non-ventilatory functions such as cough and airway clearance which is associated with increased morbidity [7].

Oxidative stress is a predominant feature of OSAS with the reactive oxygen species (ROS) produced as a result of recurrent hypoxia/reoxygenation cycles having detrimental consequences for a plethora of integrative bodily systems, highlighting a therapeutic role for antioxidants [8]. Aberrant plasticity at multiple levels of the respiratory control system has be shown to be reliant upon CIH-induced ROS [9]. CIH-induced plasticity extends to the diaphragm with diaphragm dysfunction observed following two weeks [10] and five weeks [11] of exposure to CIH in rat models. While the underlying mechanisms of CIH-induced diaphragm dysfunction are largely underexplored, antioxidants have proved efficacious in preventing diaphragm weakness and fatigue, strongly implicating a deleterious contribution of increased ROS production to diaphragm muscle dysfunction [10]. Antioxidants including tempol, apocynin, and NAC have each been shown to prevent CIH-induced diaphragm muscle fatigue. These findings are reminiscent of studies in other disease models, which demonstrate that apocynin successfully ameliorates diaphragmatic contractile dysfunction [12,13].

The contractile components of muscle fibres alter their function in response to cellular redox state, highlighting the role which ROS have in modulating skeletal muscle performance [14]. ROS are generated by cellular complexes including mitochondria, xanthine oxidase, phospholipase A2, and NADPH oxidase (NOX) [15]. The NOX family of enzymes is composed of NOX 1–5 and DUOX 1 and 2. CIH-induced NOX2-dependent effects have been demonstrated in a range of tissues including the brain, heart, and testes [16,17,18]. NOX2 subunit expression has been demonstrated in the sarcoplasm and t-tubules of skeletal muscle, highlighting the potential role of NOX2-derived ROS in alterations to the contractile apparatus of muscle fibres and thus, muscle function [19,20]. Interestingly, mouse models of Duchenne muscular dystrophy (DMD) [21] and congestive heart failure (CHF) [22] have implicated a pivotal role for NOX2 in disease-related diaphragm dysfunction.

In the current study, adult male mice were exposed to a well-established, mild-to-moderate paradigm of CIH. Pharmacological and transgenic approaches were utilised to determine if NOX2-derived ROS underpin CIH-induced dysfunction in contractile function of the diaphragm muscle. We hypothesised that exposure to CIH results in diaphragm muscle dysfunction and that blockade or deletion of NOX2 prevents CIH-induced impairment of muscle contractile function. Additionally, we hypothesised that exposure to CIH would cause NOX2-dependent transcriptional changes in genes essential for optimal muscle function.

## 2. Materials and Methods

### 2.1. Ethical Approval

Procedures on live animals were performed under license from the Government of Ireland Department of Health (B100/4498) in accordance with National and European legislation (2010/63/EU) following approval by University College Cork Animal Research Ethics Committee (AEEC no. 2013/035).

### 2.2. Chronic Intermittent Hypoxia Animal Model

C57BL/6J male mice (9 weeks old) were obtained from Envigo, UK, and assigned to one of three experimental groups: normoxic control (sham; n = 20), chronic intermittent hypoxia (CIH)-exposed (CIH; n = 20), and CIH + apocynin (CIH + APO; n = 20). Temperature and humidity-controlled rooms were used to house animals conventionally, operating on a 12 h light/12 h dark cycle. For the duration of the study, food and water was available ad libitum. Mice were housed in standard home cages placed in commercially designed hypoxia chambers for daily gas treatments (Oxycyler^™^, Biospherix, Lacona, NY, USA). Exposure to CIH consisted of intermittent cycling of gas from normoxia (21% O_2_) for 210 s to hypoxia (5% O_2_ at the nadir) over a 90 s period resulting in 12 cycles/hr for 8 hr/day. Exposures were performed during light hours for 14 consecutive days as described previously [23]. The NADPH oxidase 2 inhibitor, apocynin, was placed in the drinking water of the CIH + APO group (2 mM) for the duration of the exposure to CIH. The apocynin drug solution was freshly made and changed daily. The sham (control) group of mice was exposed to 21% O_2_ in adjacent environmental chambers during daily gas treatments. In addition, NOX2-null male mice (9 weeks old; B6.129S-Cybb^tm1Din^/J backcrossed on C57 for multiple generations) were purchased from the Jackson Laboratory (Bar Harbor, ME, USA) and assigned to a sham (NOX2 knockout (KO) sham; n = 12) or CIH (NOX2 KO CIH; n = 12) exposure in separate studies. These mice were housed in individually ventilated cages when not undergoing exposure to CIH due to the immunocompromised nature of NOX2 deficient mice. Exposure to sham or CIH was commenced at 11 weeks of age. All mice were studied the day after 14 consecutive days of exposure. Mice were weighed daily. Following 14 consecutive days of exposure, breathing was examined using whole body plethysmography (WBP) on a cohort of mice [24]. Thereafter, mice were anaesthetised with 5% isoflurane in air and euthanised humanely via cervical spinal dislocation. Sternohyoid muscles were studied and the study findings are reported elsewhere [25]. Diaphragm muscles were excised and used for ex vivo assessment of muscle function (Section 2.3) or were snap frozen in liquid nitrogen and stored at −80 °C for further tissue processing prior to gene (Section 2.4) and protein (Section 2.5, Section 2.6 and Section 2.7) analysis.

### 2.3. Ex Vivo Muscle Function Analysis

#### 2.3.1. Muscle Dissection and Preparation

The diaphragm muscle was excised in its entirety by firmly holding the central tendon and cutting around the rib cage. The whole diaphragm was subsequently placed in a storage bath of hyperoxic (95% O_2_, 5% CO_2_) Krebs solution (NaCl 120 mM, KCl 5 mM, Ca^2+^ gluconate 2.5 mM, MgSO_4_ 1.2 mM, NaH_2_PO_4_ 1.2 mM, NaHCO_3_ 25 mM, glucose 11.5 mM and d-tubocurarine 25 μM) at room temperature. The diaphragm was cut longitudinally in the direction of its muscle fibres to generate multiple uniform strips of muscle (~2 mM in diameter) including central tendon and rib. The strips of diaphragm muscle were returned to the hyperoxic Krebs solution for 10–15 min to recover from dissection. A strip of diaphragm muscle was suspended vertically between two platinum plate electrodes in a water-jacketed tissue bath at 35 °C containing Krebs solution. Preparations were continuously gassed with carbogen (95% O_2_ and 5% CO_2_). Using non-elastic string, a strip of diaphragm muscle was attached to the immobile hook at the rib end and to the dual-mode lever transducer at the central tendon (Aurora Scientific Inc.; Aurora, ON, Canada) with platinum plate electrodes flanking both sides of the muscle. The experimental protocol began after the muscle had a 15 min equilibration period in the muscle bath.

#### 2.3.2. Isometric Muscle Function

Diaphragm muscle function was examined utilising various protocols previously described [26,27,28,29,30]. Diaphragm muscles from all groups (sham, CIH, CIH + APO, NOX2 KO sham, and NOX2 KO CIH; n = 8–10 per group) were examined. In order to assess isometric muscle contractions, the force transducer was set to maximum tension (>100% load) for the duration of these assessments. Diaphragm muscle preparations were stimulated at supramaximal voltage via the platinum plate electrodes flanking both sides of the muscle strip. By stimulating the muscle supra-maximally for 1ms, the optimal length (L_o_), the length that produces the maximum twitch force, was established for each strip. Using a micro-positioner, the length of the muscle was adjusted between each of these stimulations and once the L_o_ was obtained, the muscle remained at this length for the duration of the protocol. Twitch force and various contractile kinetics including contraction time (CT; time to peak force) and half-relaxation time (1/2 RT; time for force to decay by 50%) were determined from a single twitch stimulation. Subsequently, an isometric tetanic contraction was evoked by supramaximal stimulation at a frequency of 100 Hz for 300 ms to determine the peak tetanic force (F_max_). The CSA of each strip was determined by dividing the muscle mass (weight in grams) by the product of muscle L_o_ (cm) and muscle density (assumed to be 1.06 g/cm^3^). Specific force was calculated in N/cm^2^ of estimated muscle cross sectional area (CSA).

#### 2.3.3. Isotonic Muscle Function

The force transducer was set to varying degrees of tension (0–100% load) for the duration of the isotonic protocol. Initially, the force transducer was set to minimum tension (0%) and the muscle was stimulated to contract against a load that equated to 0% of its Fmax, previously determined. This load was subsequently increased in a step-wise manner (5%, 10%, 20%, 30%, 40%, 60%, 80%, 100%) % of Fmax. At each step, a contraction was elicited. Notably, 30 s was allowed between each contraction to allow the muscle to fully return to L_o_ before the next contraction was elicited. The length of shortening was defined as the maximum distance of shortening during the contraction. Shortening velocity was determined from the distance of shortening during the first 30ms of shortening during a contraction. Peak shortening length and velocity were achieved at 0% of Fmax. Peak specific shortening (Smax) was defined as the length of shortening per optimal length (L/L_o_). Peak specific shortening velocity (Vmax) was defined as L_o_/s. We calculated the work (force x shortening) and power-generating capacity (force x shortening velocity) of the diaphragm muscle at each load. Work-load and power-load relationships were determined as well as peak work and peak power. Peak specific mechanical work (Wmax) was calculated as Joules/cm^2^ and peak specific mechanical power (Pmax) as Watts/cm^2^.

### 2.4. Quantitative Reverse Transcription Polymerase Chain Reaction (qRT-PCR)

#### 2.4.1. RNA Extraction and Preparation

Diaphragm muscle samples from each experimental group (sham, CIH, CIH + APO, NOX2 KO sham, and NOX2 KO CIH; n = 6–9 per group) were examined. Samples were weighed following removal from storage at −80 °C. A general lab homogeniser (Omni-Inc., Kennesaw, GA, USA) was used to homogenise samples (20–40 mg) in Tripure Isolating reagent (Roche Diagnostics Ltd., West Sussex, UK) kept on ice. The resultant homogenates were placed on ice for 20 min with intermittent vortexing to promote cell lysis. The phenol-chloroform method was utilised to extract total RNA from homogenates, in accordance with the manufacturer’s instructions. For muscle samples, an additional chloroform wash step was performed to enhance the purity of isolated RNA. By spectrophotometry, a nanodrop 1000 (Thermo Scientific, Waltham, MA, USA) was used to assess both the quantity (ng/µL) and purity of isolated RNA (260:280 and 260:230 ratios). Using an agarose gel electrophoresis system (E-gel, Life Technologies, Carlsbad, CA, USA), the integrity of isolated RNA was assessed by visualization of distinct 18S and 28S ribosomal RNA bands.

#### 2.4.2. cDNA Synthesis

Diaphragm muscle RNA was reverse transcribed to cDNA using a Transcriptor First Strand cDNA Synthesis Kit (Roche Diagnostics Ltd., Burgess Hill, UK) carefully following the manufacturer’s instructions.

#### 2.4.3. qRT-PCR

Realtime ready Catalog or Custom assays (Roche Diagnostics Ltd., Burgess Hill, UK) shown in Table 1 and Fast Start Essential Probe Master (Roche Diagnostics Ltd., Burgess Hill, UK) were used to amplify cDNA, as per the manufacturer’s instructions. Briefly, each reaction consisted of five µL cDNA and fifteen µL master mix. All reactions were carried out in duplicate on a 96-well plate using the Lightcycler 96 for the protocol (Roche Diagnostics Ltd., Burgess Hill, UK). Several negative controls including RNA negatives, reverse transcription negatives, cDNA negatives (no template controls) and plate calibrators were used on every plate to ensure specific amplification. Quantification cycle values (Cq) extracted from experiments for the genes of interest (Table 1) were normalized to that of a reference gene, *Hprt1*, to account for variations in input amounts of RNA/cDNA and the efficiency of reverse transcription. Following a screen of potential candidate reference genes, *Hprt1* was found to be the most stable gene when gas exposure (hypoxia), drug treatment (apocynin), and genotype (NOX2 null) were considered. The relative gene expression was calculated using the ∆∆CT method (normalised expression of the gene of interest to that of the reference gene), with a change in expression shown as a fold change relative to the sham exposure (control group).

### 2.5. Western Blotting

#### 2.5.1. Protein Extraction and Quantification

Diaphragm muscles were removed from storage and weighed immediately. Muscle samples from sham and CIH-exposed groups were homogenised in modified ice cold radioimmunoprecipitation assay buffer (RIPA buffer) containing: RIPA (25 mM Tris-HCl pH 7.6, 150 mM sodium chloride, 1% NP-40, 1% sodium deoxycholate, 0.1% SDS), deionised water, protease inhibitor cocktail (104 mM AEBSF, 80 µM aprotinin, 4 mM bestatin, 1.4 mM E-64, 2 mM leupeptin, 1.5 mM pepstatin A), phosphatase inhibitor cocktail (200 mM sodium fluoride, 5 mM sodium orthovanadate) and 100 mM PMSF (Sigma Aldrich, Arklow, Wicklow, Ireland) using a 10% *w*/*v* ratio with 8 × 10 s bursts using a general laboratory homogeniser (Omni-Inc., Kennesaw, GA, USA). To promote cell lysis, homogenates were placed on ice for 20 min with vortexing performed at 4 min intervals. Samples were centrifuged (U-320R centrifuge Boeckel + Co, Hamburg, Germany) for 20 min at 4 °C at 15,366× *g*. The supernatant was removed and stored at −80 °C for further use. A bicinchoninic acid (BCA) protein quantification assay (Pierce Biotechnology, Dublin, Ireland) was used to determine the protein concentration of each sample, carefully following the manufacturer’s instructions. Absorbance was measured at 562nm using a SpectraMax-M3 spectrophotometer (Molecular Devices, Sunnyvale, CA, USA).

#### 2.5.2. Gel Electrophoresis

Western blot experiments were performed on diaphragm muscle homogenates from sham and CIH-exposed mice (n = 5–7 per group). Samples were appropriately diluted based on their protein concentration and combined with an equal volume of 2X SDS-PAGE loading buffer (4% SDS (sodium dodecyl sulfate), 200 mM DTT (dithiothreitol), 20% glycerol, 100 mM TRIS-CL (pH 6.8), 0.2% Bromophenol blue). Following this, samples underwent boiling at 100 °C for 5 min on a dry heating block (Techne, Abingdon, UK). Fifteen μg of protein from each sample was resolved on 10% SDS-polyacrylamide gels (deionised water, 30% acrylamide, 1.5M TRIS pH 8.8, 10% SDS, 10% a mMonium persulfate). Proteins resolved on the gel were then transferred onto nitrocellulose membranes electrophoretically using a semi-dry transfer rig (BioRad, Hercules, CA, USA). Membranes were incubated in 0.1% (*w*/*v*) Ponceau S in 5% acetic acid to reversibly stain the transferred proteins to assess equal protein loading and transfer. Membranes were digitally photographed for densitometric analysis for normalisation purposes. Membranes were blocked for 1 h in TBST (20 mM Tris-HCl, pH 7.6, 150 mM NaCl, 0.1% Tween) containing 5% non-fat dried milk and were incubated overnight with the primary antibody specific for the protein of interest as follows: anti-NOX2, anti-NOX4; 1:10,000; 1:2000 (Abcam, Cambridge, UK) in 5% bovine serum albumin (BSA)/5% non-fat dried milk. The following day, membranes were incubated for 1 h at room temperature with a 1:2000 dilution of HRP-linked anti-mouse secondary antibody (Cell Signaling Technology, Danvers, MA, USA) in 5% non-fat dried milk/TBST. Enhanced chemiluminescence reagent (ECL Plus, GE Healthcare, Chalfont Saint Giles, UK) and exposure to chemiluminescent sensitive film (Kodak, Rochester, NY, USA) were used in combination to visualise bands. Films were developed, digitally photographed and densitometric analysis of bands of interest was performed (QuantityOne, Biorad, Naas, Kildare, Ireland). Band intensities of proteins of interest were normalised to the intensities of the corresponding Ponceau S staining proteins. This was performed to adjust for protein loading, thus allowing comparative analysis between sham and CIH-exposed diaphragm muscles. Values are expressed as optical density (O.D)/Ponceau S (arbitrary units; a.u).

### 2.6. Spectrophotometric Assays

#### 2.6.1. Protein Extraction and Quantification

Diaphragm muscles were homogenised, and protein was quantified using the protocol previously described in Section 2.5.1.

#### 2.6.2. NADPH Oxidase Activity

NADPH oxidase (NOX) enzyme activity was measured in diaphragm muscle homogenates from CIH-exposed and sham mice (n = 8 per group). Experiments were performed using a cocktail mixture consisting of: nitroblue tetrazolium (NTB, 2.2 mM in water), Tris-HCL pH8 (2.8 mM), and diethylene-triamine-penta-acetic acid (1.3 mM in Tris-HCL) and a fresh solution of NADPH (1 mM). The reaction mixture contained 20 μL of muscle homogenate, 250 μL of cocktail mixture and 30 μL of NADPH. This solution was added in duplicate to the wells in a black 96-well plate. The 96-well plate was gently shaken at room temperature for 2 min. Absorbance was measured at 1 min intervals over a period of 30 min in a SpectraMax-M3 spectrophotometer (Molecular Devices, Sunnyvale, CA, USA) at 560 nm. NOX activity was calculated based on the slope of the formation of formazan blue over time. Data were expressed as enzymatic activity per mg of protein in the sample (previously determined by BCA assay), per minute.

#### 2.6.3. Citrate Synthase Activity

Citrate synthase is the initial enzyme of the tricarboxylic acid (TCA) cycle and is routinely used as an indicator of mitochondrial integrity. Citrate synthase activity was assessed in diaphragm muscle homogenates from CIH-exposed and sham mice (n = 8 per group). The citrate synthase activity assay was performed in accordance with the manufacturer’s instructions (CS0720; Sigma-Aldrich, Arklow, Wicklow, Ireland). Data were expressed as enzymatic activity per μmole per min per mg protein.

#### 2.6.4. Thiobarbituric Acid Reactive Substances

Thiobarbituric acid reactive substances (TBARS) are commonly used as a marker of lipid peroxidation, an indirect assessment of oxidative stress. Levels of TBARS were examined in diaphragm muscle homogenates from sham and CIH-exposed mice (n = 8 per group). A standard curve was curated using malondialdehyde (MDA). Fifty μL of thiobarbituric acid (TBA, 50  mM) was added to 50 μL of diaphragm muscle homogenate. The solution was then incubated at 97 °C for 1 h on a dry heating block. Samples were immediately cooled on ice, and 75 μL of methanol: 1  mM NaOH (91:9) was added to the mixture. Samples centrifuged at 704× *g*. 70 μL of the resultant supernatant was added in duplicate per well in a black 96-well plate. A spectraMax-M3 spectrophotometer (Molecular Devices, USA) using 523/553 excitation/emission settings as used to read the 96-well plate. Data are expressed as nM TBARS per mg of protein.

### 2.7. Cell Signalling Assays

#### 2.7.1. Protein Extraction and Quantification

Diaphragm muscles were homogenised and prepared as described in Section 2.5.1. To optimise samples for cell signalling assays some optimisations were required. A 2.5% *w*/*v* ratio of tissue to RIPA buffer, twice the amount of phosphatase inhibitor and a 1:3 dilution of homogenates for the BCA assay were used.

#### 2.7.2. Hypertrophy, Atrophy, and HIF Signalling Assays

Cell signalling assays were carried out using a PhosphoFOXO3a; a MAP Kinase phosphoprotein panel–phosphop-38, phospho-ERK1/2, and phospho-JNK; and total HIF-1α assay (MesoDiscovery, Rockville, MD, USA). Cell signaling assays were performed on diaphragm muscle homogenates (n = 8 per group) from CIH-exposed or sham mice in accordance with the manufacturer’s instructions (MesoDiscovery, USA). Values are expressed as signal/μg of total protein.

### 2.8. Statistical Analysis

Prism 8.0 (Graphpad Software, San Diego, CA, USA) was used to statistically compare and graphically display data. Data are shown in violin plots (individual data points with median and interquartile range) or reported as mean ± SD. Two statistical approaches were taken in the current study. For comparisons in wild-type groups (sham, CIH, and CIH + APO), one-way ANOVA with Tukey’s post hoc test was used to compare data sets with confirmed normal distribution. A non-parametric Kruskal–Wallis with Dunn’s post hoc test was used to compare data sets, which were not normally distributed. All *p* values are adjusted to account for multiple comparisons. *p* < 0.05 was deemed statistically significant. For comparisons in KO groups (NOX2 KO sham and NOX2 KO CIH), an unpaired two-tailed Student’s *t* test was used to compare data sets, which were normally distributed and of confirmed equal variance. In the case of unequal variance, Welch’s correction was applied. A non-parametric Mann–Whitney test was used to compare data, which were not normally distributed. *p* < 0.05 was deemed statistically significant. Repeated measures two-way ANOVA (RMANOVA) with Bonferroni post hoc test was used to statistically compare isotonic measures of diaphragm muscle contractile performance across all groups (Figure 1). Statistical significance was taken at *p* < 0.05.

## 3. Results

### 3.1. Diaphragm Muscle Contractile Function Ex Vivo

Diaphragm twitch kinetics (P_t_, CT and ½ RT) and isotonic contractile parameters (Pmax, Wmax, Smax, and Vmax) are shown in Table 2 and Table 3. Two weeks of exposure to CIH had no effect on diaphragm twitch force (P_t_) compared to wild-type sham mice (*p* = 0.2738; Table 2); administration of apocynin (2 mM) in the drinking water throughout the exposure to CIH significantly increased twitch force compared with exposure to CIH alone (*p* = 0.0016; Table 2). P_t_ was unaffected in NOX2 KO mice exposed to CIH compared to NOX2 KO sham mice (*p* = 0.1107; Table 3). Twitch kinetics including contraction time (CT) and half-relaxation time (½ RT) were unchanged following exposure to CIH in both wild-type (Table 2) and NOX2 KO mice (Table 3). Figure 1 shows representative original traces for (A) sham (grey), CIH (black), CIH + APO (blue), and (B) NOX2 KO sham (pink) and NOX2 KO CIH (red) tetanic contractions. Exposure to CIH for two weeks significantly decreased the force-generating capacity of the diaphragm compared to wild-type sham mice (*p* = 0.0042; Figure 1C). Compared to sham mice, exposure to CIH resulted in a ~45% decrease in peak specific force of the diaphragm. Apocynin administration in the drinking water throughout the CIH exposure entirely prevented CIH-induced muscle weakness in wild-type mice (*p* = 0.0015; Figure 1C). Peak tetanic force produced by NOX2 KO mice exposed to CIH was unchanged compared to NOX2 KO sham controls (*p* = 0.4613; Figure 1D). Exposure to CIH had no effect on diaphragm muscle mass compared to sham both in wild-type (Table 2) and NOX2 KO mice (Table 3). Body mass was significantly decreased following exposure to CIH in both wild-type (*p* = 0.0032; Table 2) and NOX2 KO mice (*p* = 0.0042; Table 3). Apocynin treatment throughout the CIH exposure prevented the CIH-induced decrease in mouse body mass (*p* = 0.0047; Table 2).

Representative original traces of ex vivo diaphragm muscle maximum unloaded shortening for (E) sham (grey), CIH (black), CIH + APO (blue), (F) NOX2 KO sham (pink), and NOX2 KO CIH (red) preparations are shown in Figure 1. Figure 1G,H, show the diaphragm power-load relationships of the experimental groups. The power-generating capacity of the diaphragm was significantly decreased following exposure to CIH (*p* < 0.0001; Figure 1G). Post hoc analysis showed a statistically significant decrease in the power-generating capacity of the diaphragm at 20% (* *p* < 0.05), 30% (** *p* < 0.01), 40% (** *p* < 0.01), and 60% (* *p* < 0.05) of its peak force-generating capacity following exposure to CIH, when compared to wild-type sham mice (Figure 1G). Administration of apocynin throughout the exposure to CIH ameliorated the CIH-induced decrease in the power-generating capacity of the diaphragm compared to mice exposed to CIH alone (*p* < 0.0001; Figure 1G). A statistically significant increase in the power-generating capacity of the diaphragm following treatment with apocynin throughout exposure to CIH at 10–60% of the peak force-generating capacity compared to CIH-exposed mice was revealed following post hoc analysis (Figure 1G). Two weeks of CIH exposure significantly decreased peak power (Pmax) compared to wild-type sham mice (*p* = 0.0193; Table 2); Apocynin treatment prevented the decrease in Pmax compared to CIH exposure alone (*p* = 0.0392; Table 2). The power-generating capacity of the diaphragm in NOX2 KO mice over the range of loads assessed was unaffected by exposure to CIH when compared to NOX2 KO sham controls (Figure 1H). Exposure to CIH had no effect on Pmax in NOX2 KO mice (*p* = 0.1821; Table 3). Figure 1I,J, show the diaphragm work-load relationship across all groups. The work produced by the diaphragm muscle over a range of loads was significantly decreased following exposure to CIH compared to wild-type sham mice (*p* < 0.0001; Figure 1I). A statistically significant decrease in mechanical work of the diaphragm following exposure to CIH at 20% (* *p* < 0.05), 30% (*** *p* < 0.001), 40% (**** *p* < 0.0001), and 60% (**** *p* < 0.0001) of the peak force-generating capacity compared to wild-type sham mice was revealed following post hoc analysis (Figure 1I). The administration of apocynin throughout the CIH exposure ameliorated the CIH-induced decrease in the work produced by the diaphragm over a range of loads compared to mice exposed to CIH alone (*p* < 0.0001; Figure 1I). A statistically significant increase in work produced by the diaphragm following treatment with apocynin throughout CIH exposure at 20% (^###^
*p* < 0.001), 30% (^####^
*p* < 0.0001), 40% (^####^
*p* < 0.0001), and 60% (^###^
*p* < 0.001) of the peak force-generating capacity compared to CIH-exposed mice was revealed following post hoc analysis (Figure 1I). Peak work (Wmax) was significantly reduced following exposure to CIH compared to wild-type sham mice (*p* = 0.0060; Table 2); apocynin administration completely prevented the CIH-induced decrease in Wmax (*p* = 0.0030; Table 2). The work produced by the diaphragm in NOX2 KO mice over the range of loads examined was unaffected by two weeks of CIH exposure compared to NOX2 KO sham controls (Figure 1J). Similarly, exposure to CIH had no effect on Wmax in NOX2 KO mice (*p* = 0.3923; Table 3).

Representative original traces of ex vivo diaphragm muscle maximum unloaded shortening for (K) sham (grey), CIH (black), CIH + APO (blue), (L) NOX2 KO sham (pink), and NOX2 KO CIH (red) preparations are shown in Figure 1. The diaphragm shortening-load relationship across all groups is shown in Figure 1M,N. Exposure to CIH for two weeks decreased the distance of shortening of the diaphragm muscle when compared to wild-type sham mice (*p* < 0.0001; Figure 1M); apocynin treatment significantly increased the distance of shortening of the diaphragm over a range of loads examined compared with exposure to CIH alone (*p* < 0.0001; Figure 1M). Exposure to CIH or apocynin treatment throughout the CIH exposure had no effect on peak shortening (Smax) in wild-type mice (*p* = 0.1374; Table 2). The distance of shortening over the range of loads examined was unaffected by exposure to CIH in NOX2 KO mice compared to sham controls (*p* = 0.3893; Figure 1N). Exposure to CIH had no effect on Smax in NOX2 KO mice (*p* = 0.1148; Table 3). Figure 1O,P show the diaphragm shortening velocity-load relationship across all groups. Exposure to CIH decreased the shortening velocity of the diaphragm muscle compared to wild-type sham mice (*p* = 0.0006; Figure 1O); apocynin treatment prevented the decrease in the shortening velocity of the diaphragm over a range of loads examined compared with exposure to CIH alone (*p* = 0.0123; Figure 1O). Exposure to CIH or treatment with apocynin throughout the CIH exposure had no effect on the peak shortening velocity (Vmax) in wild-type mice (*p* = 0.1403; Table 2). Exposure to CIH in NOX2 KO mice had no effect on shortening velocity. Exposure to CIH had no effect on Vmax in NOX2 KO mice (*p* = 0.3958; Table 3).

### 3.2. NOX mRNA and Protein Expression in Diaphragm Muscle

The mRNA expression of the most predominant NOX isoforms in naïve wild-type mouse diaphragm muscle is shown in Figure 2A. There was a significantly lower mRNA expression of NOX1 compared with both NOX4 (*p* < 0.0001; Figure 2A) and NOX2 (*p* < 0.0001; Figure 2A) in the diaphragm muscle. The mRNA expression of NOX4 compared with NOX2 in diaphragm muscle was not different (*p* = 0.9709; Figure 2A). Data for NOX2 mRNA and protein expression in mouse diaphragm muscle are shown in Figure 2B–D. The mRNA expression of NOX2 was unaltered following CIH exposure compared with sham in wild-type mice; NOX2 mRNA levels were significantly decreased in the diaphragm following apocynin co-treatment throughout the CIH exposure compared with exposure to CIH alone (*p* = 0.0323; Figure 2B). An absence of NOX2 mRNA was confirmed in NOX2 KO diaphragm muscle (Figure 2C). Western blot bands were detected at approximately 65 kDa (predicted molecular weight (MW) of the NOX2 subunit) (Figure 2D). The corresponding Ponceau S-stained membrane is shown in Figure 2D, confirming relatively equal protein content per well. Exposure to CIH had no effect on the protein expression of the NOX2 subunit in the diaphragm of mice compared with sham in wild-type mice (*p* = 0.3806; Figure 2D). Data for NOX4 mRNA and protein expression in mouse diaphragm muscle are shown in Figure 2E–G. The mRNA expression of NOX4 was significantly increased following two weeks of exposure to CIH compared with sham in wild-type mice (*p* = 0.0331; Figure 2E); apocynin treatment prevented the CIH-induced increase in NOX4 mRNA levels (*p* = 0.0011; Figure 2E). Additionally, NOX4 mRNA expression was significantly increased in the diaphragm of NOX2 KO mice following CIH exposure (*p* = 0.0040; Figure 2F). Western blot bands were detected at approximately 63 kDa (predicted MW of the NOX4 subunit) (Figure 2G). The corresponding Ponceau S-stained membrane is shown in Figure 2G. The protein expression of the NOX4 subunit was significantly increased in the diaphragm of mice exposed to CIH compared to sham in wild-type mice (*p* = 0.0014; Figure 2G). Diaphragm muscle mRNA expression of NOX catalytic and accessory subunits are shown in Figure 2 h–W. The mRNA expression of p22phox was decreased in the diaphragm muscle of mice exposed to CIH compared to sham controls, but this did not meet the threshold for statistical significance (*p* = 0.0840; Figure 2J); apocynin treatment significantly increased p22phox mRNA expression compared with exposure to CIH alone (*p* = 0.0113; Figure 2J). CIH exposure had no effect on the mRNA expression of p22phox in the diaphragm of NOX2 KO mice (*p* = 0.8102; Figure 2K). There were no statistically significant differences in the mRNA expression of NOX1 (Figure 2 h,I), p47phox (Figure 2L,M), p40phox (Figure 2N,O), p67phox (Figure 2P,Q), Rac (Figure 2R,S), DUOX1 (Figure 2T,U), or DUOX2 (Figure 2V,W) across all groups. Exposure to CIH for two weeks did not affect NOX enzymatic activity in diaphragm muscle compared with sham (*p* = 0.5565; Table 4).

### 3.3. Indices of Redox Balance, Protein Synthesis and Degradation in Diaphragm Homogenates from Sham and CIH-Exposed Mice

Assessments of cell signalling in the diaphragm of sham and CIH-exposed wild-type mice are shown in Table 4. Levels of TBARS (marker of lipid peroxidation) were unaffected by exposure to CIH in the diaphragm muscle compared to sham controls (*p* = 0.3536; Table 4). Similarly, citrate synthase activity (*p* = 0.5892; Table 4), HIF-1α (*p* = 0.5982; Table 4) protein content or Phospho-FOXO-3α (*p* = 0.3284; Table 4) were unchanged compared to sham. The expression of key proteins involved in the MAPK signalling pathway (*p* = 0.0379, Phopho-ERK1/2; *p* = 0.0287, Phospho-JNK; *p* = 0.0007; Phospho-p38) were significantly decreased in diaphragm muscle following exposure to CIH when compared to sham controls.

### 3.4. mRNA Expression of Genes Relating to Myogenesis in Diaphragm Muscle

The mRNA expression of various genes involved in myogenesis in diaphragm muscle across all groups is shown in Figure 3. Exposure to CIH significantly increased diaphragm mRNA expression of Myostatin (*p* = 0.0103; Figure 3A), MyoD (*p* = 0.0045; Figure 3C), Myogenin (*p* = 0.0035; Figure 3E), MEF2C (*p* = 0.0100; Figure 3G) and Sirtuin-1 (*p* = 0.0282; Figure 3I). Apocynin treatment ameliorated the CIH-induced increase in MyoD (*p* < 0.0001; Figure 3C), Myogenin (*p* < 0.0001; Figure 3E), MEF2C (*p* = 0.0003; Figure 3G), and Sirtuin-1 (*p* = 0.0004; Figure 3I). The mRNA expression genes involved in myogenesis were unaffected following exposure to two weeks of CIH in NOX2 KO mice (Figure 3B,D,F,H,J,L). The mRNA expression of IGF-1 (Figure 3K,L) was unchanged across all groups. Heat maps summarising the diaphragm mRNA expression data for these genes across all groups examined are shown in Figure 3M,N.

### 3.5. mRNA Expression of Genes Relating to Antioxidant Status in Diaphragm Muscle

The mRNA expression of genes contributing to antioxidant status in diaphragm muscle across all groups is shown in Figure 4. The mRNA expression of SOD1 (Figure 4A), SOD2 (Figure 4C), and Catalase (Figure 4E) was unaffected by CIH exposure in wild-type mice compared to sham controls. Similarly, apocynin treatment had no effect on the mRNA expression of SOD1 (Figure 4A), SOD2 (Figure 4C), or Catalase (Figure 4E) in wild-type mice compared with exposure to CIH alone. The mRNA expression of SOD1 (*p* = 0.0150; Figure 4B) and Catalase (*p* = 0.0174; Figure 4F) was significantly elevated following CIH exposure in NOX2 KO mice. The mRNA expression of NRF2 (*p* = 0.0016; Figure 4G) was significantly increased following CIH exposure in wild-type mice compared to sham controls; apocynin administration throughout the CIH exposure ameliorated this increase (*p* < 0.0001; Figure 4G). The mRNA expression of NRF2 (Figure 4G) or SOD2 (Figure 4D) was unaltered following CIH exposure in NOX2 KO mice. Heat maps summarising the diaphragm mRNA expression data for these genes across all groups examined are shown in Figure 4I,J.

### 3.6. mRNA Expression of Genes Relating to Inflammation and Protein Degradation in Diaphragm Muscle

The mRNA expression of genes involved in inflammatory and protein degradation processes in diaphragm muscle across all groups is shown in Figure 5. The mRNA expression of NF-κB was elevated following CIH exposure compared to wild-type sham mice, but this did not meet the threshold for statistical significance (*p* = 0.0696; Figure 5A); apocynin treatment significantly prevented this increase (*p* = 0.0006; Figure 5A). The mRNA expression of Atrogin-1was unaffected by CIH exposure in wild-type sham mice (*p* = 0.2917; Figure 5C); apocynin treatment significantly decreased the mRNA expression of Atrogin-1 compared to exposure to CIH alone (*p* = 0.0137; Figure 5C). The mRNA expression of MuRF-1 (*p* = 0.0253; Figure 5E), PARK-2 (*p* = 0.0116; Figure 5I). BNIP-3 (*p* = 0.0189; Figure 5K) and LC3B (*p* = 0.0244; Figure 5; M) was significantly elevated following two weeks of CIH exposure in wild-type sham mice. Apocynin treatment significantly ameliorated each of these increases compared to exposure to CIH alone (MuRF-1; *p* = 0.0010; Figure 5E, PARK-2; *p* = 0.0003; Figure 5I, BNIP-3; *p* = 0.0006; Figure 5K, LC3B; *p* = 0.0048; Figure 5M, respectively). The mRNA expression of NF-κB (Figure 5B), Atrogin-1 (Figure 5D), MuRF-1 (Figure 5F), PARK-2 (Figure 5J), BNIP-3 (Figure 5L), or LC3B (Figure 5N) was unaltered by CIH exposure in NOX2 KO mice. There were no alterations in the mRNA expression of PINK-1 (Figure 5G,H) or GABARAPL1 (Figure 5O,P) across all groups. Heat maps summarising the diaphragm mRNA expression data for these genes across all groups examined are shown in Figure 5Q,R.

## 4. Discussion

The diaphragm muscle elaborates a mixed fibre type composition, enabling its impressive adaptability and plasticity characteristics. OSAS patients [4,5], and rodents subjected to recurrent tracheal occlusion [31] show evidence of diaphragm muscle maladaptation. However, there appears to be significant heterogeneity in the respiratory presentation of OSAS as evidence also exists to support an absence of diaphragm dysfunction in OSAS patients [32]. Ordinarily, the upper airway remains patent due to the pivotal balance between the sub-atmospheric collapsing pressure of the diaphragm and the dilating forces produced by upper airway muscles protecting airway calibre [6]. The diaphragm muscle is under considerable strain in OSAS due to heavy and prolonged contractions during hypoxic periods, characteristic of this disease. The resultant tissue hypoxia and disordered energy balance may serve to predispose the diaphragm muscle to further stress and dysfunction [33]. Evidence suggests that the diaphragm is also susceptible to fatigue during apnoeic episodes in OSAS patients [34]. Diaphragm fatigue may be protective in OSAS by reducing the collapsing force during apnoeic episodes. This in turn would counterbalance the consequences of upper airway muscle dysfunction in OSAS. In contrast, diaphragm muscle dysfunction may reduce the ability to generate large powerful manoeuvres necessary for non-ventilatory functions such as airway clearance and cough. CIH-induced diaphragm muscle plasticity has been demonstrated; however, the underlying mechanisms are largely unknown.

The effects of exposure to CIH are hugely dependent on the duration and intensity of the stimulus and as a result, evidence shows a variety of effects of IH on diaphragm muscle [10,11,33,35,36]. In the present study, exposure to a mild-to-moderate paradigm of CIH for two weeks significantly reduced the peak force-generating capacity of the diaphragm by nearly half compared with sham controls. This finding is consistent with diaphragm weakness and/or fatigue demonstrated following two weeks [10] and five weeks [11] of exposure to CIH in rat models. Interestingly, the magnitude of diaphragm weakness observed in the present study is larger than that previously observed in rat models following exposure to CIH for two weeks [10]. The diaphragm muscle weakness observed in the present study is reminiscent of diaphragm weakness observed following six weeks of exposure to chronic sustained hypoxia [37], associated with profound oxidative stress. We postulate that if the reduced power-generating capacity of the diaphragm over physiological loads following CIH exposure we observed translates to the in vivo setting, then it would hold significance for human OSAS as well as other diseases defined by exposure to CIH.

The contractile kinetics (CT and ½ RT) of the diaphragm were unaltered following two weeks of exposure to CIH when compared with sham controls, consistent with previous studies in rat models exposed to varying paradigms and lengths of CIH [11,38]. These findings suggest that Ca^2+^ handling in the diaphragm muscle is unaffected by CIH exposure, thus dysregulated SR functioning is unlikely to account for CIH-induced diaphragm weakness. Maladaptive changes in the distance of shortening or velocity of shortening of respiratory muscles can be indicative of poor mechanical efficiency, culminating in muscle dysfunction [39,40]. Exposure to CIH decreased the distance and velocity of shortening of the diaphragm muscle across a variety of physiological loads in the present study. These results are indicative of an increased load on the muscle, thereby contributing to a reduced capacity to contract and produce optimal force. We posit that the reduction in the shortening of the diaphragm may inhibit or reduce the ability of the diaphragm muscle to contract optimally, highlighting a candidate mechanism underlying CIH-induced diaphragm muscle dysfunction.

Previous studies suggest that under basal conditions, ROS exert a tonic inhibition on upper airway muscle force production [41,42,43]. Consistent with this, we have demonstrated that NOX2 is likely the dominant source of ROS driving CIH-induced sternohyoid muscle weakness [25]. We demonstrate that NOX2 knock-out results in a ~47% increase in the peak force-generating capacity of the diaphragm compared to wild-type sham mice, again revealing the powerful inhibitory influence of NOX2-dependent ROS on muscle force under basal conditions. Similarly, it is likely that apocynin has the capacity to increase muscle force per se, that is, it exerts a positive inotropic effect due to the disinhibition of basal ROS-dependent suppression of force in addition to a protective effect in preventing enhanced NOX2-derived muscle weakness (caused by CIH). Antioxidants have shown promise in ameliorating diaphragm muscle weakness and fatigue, directly implicating the overproduction of ROS in diaphragm muscle dysfunction [10]. The antioxidants tempol, apocynin, and NAC each prevented CIH-induced diaphragm muscle fatigue, with NAC showing particular promise since it prevents CIH-induced muscle weakness. In the present study, the co-treatment of mice with the NOX2 inhibitor, apocynin, throughout the exposure to CIH, entirely prevented diaphragm muscle weakness. These findings are consistent with previous studies in other models, which demonstrate that apocynin successfully ameliorates diaphragmatic contractile dysfunction [10,12,13]. Moreover, the specific role of NOX2 in CIH-induced diaphragm muscle weakness in the present study was confirmed by utilising transgenic NOX2 KO mice. The striking diaphragm muscle weakness observed in wild-type mice following exposure to two weeks of CIH was entirely absent in NOX2 KO mice, confirming the obligatory role of NOX2-dependent ROS in diaphragm muscle weakness following exposure to CIH. Our data are consistent with studies in mouse models of DMD [21] and CHF [22], which also implicate a central role for NOX2-dependent diaphragm dysfunction. Our results may have implications for human OSAS as well as other disorders characterised by oxidative stress and diaphragm muscle weakness.

ROS are capable of modulating skeletal muscle function due to the high sensitivity of contractile apparatus of muscle fibres to alterations in cellular redox state [14]. NOX1, 2 and 4 have been shown to be the most prevalent isoforms expressed in skeletal muscle cells [44,45,46,47,48]. The relative abundance of these isoforms, based on mRNA data in C2C12 cells, concludes that NOX4 is the most highly expressed NOX isoform, followed by NOX2 and subsequently NOX1 [45,47]. In line with this, we have demonstrated the presence of mRNA of NOX1, 2 and 4, as well as their respective accessory subunits (p22phox, p40phox, p47phox, p67phox, Rac), in diaphragm muscle from wild-type mice. We report no difference between the mRNA expression of NOX2 and NOX4 in the diaphragm, however we observed significantly lower levels of NOX1 mRNA. Importantly, NOX2- and NOX4-dependent ROS have been strongly implicated in muscle dysfunction in a variety of disease states [22,43,49,50,51,52,53,54]. Therefore, we hypothesize that NOX2 and NOX4 isoforms hold the most relevance for diaphragm muscle dysfunction in our rodent model.

Of interest, CIH-induced increases in NOX enzyme expression and/or activity have been reported in a variety of tissues [16,17,43,55,56,57,58,59,60,61]. In the present study, we report no alteration to the mRNA or protein expression of NOX2 in the diaphragm muscle following exposure to two weeks of CIH. While transcriptional regulation of NOX2 is widely observed, NOX2 is mainly acutely regulated through post-translational mechanisms such as the phosphorylation of NOX2’s regulatory cytosolic subunits [62]. Under basal conditions, NOX2 and p22phox are complexed in the plasma membrane with accessory subunits unbound in the cytosol. Following stimulation by factors such as hypoxia or mechanical stress, including that of muscle contraction [63], cytosolic subunits are phosphorylated and translocated to the membrane resulting in the production of superoxide through the oxidation of NADPH. Pharmacological blockade and genetic knock out of NOX2 have been shown to ameliorate diaphragm muscle weakness in models of DMD [21] and CHF [22], implicating a role for NOX2 derived ROS in muscle dysfunction. The pattern, duration, and intensity of IH is pivotal in determining a phenotypical response and as such, we suggest that the paradigm of CIH used in this study was not sufficient to induce a transcriptional change in NOX2.

Interestingly, we report an increase in NOX4 mRNA in the diaphragm of both wild-type and NOX2 KO mice following exposure to two weeks of CIH. Consistent with this, the protein expression of NOX4 was also significantly increased following CIH exposure in the diaphragm of wild-type mice. The NOX4 isoform is constitutively active, with its activity largely dependent on its level of protein expression. Therefore, the ROS that NOX4 generates are largely regulated at the gene transcription level rather than through acute, post-translational mechanisms, characteristic of NOX2 [64]. It has been suggested that NOX4 has an important role as an O_2_ sensor, with numerous studies reporting an increase in NOX4 mRNA and protein expression following hypoxia exposure [57,65]. Increased NOX4 expression in various models of diseases characterised by skeletal muscle weakness have highlighted the putative role of NOX4-dependent ROS in skeletal muscle dysfunction [66,67,68]. Additionally, CIH-induced NOX4 expression has been demonstrated in lung tissue [57]. We speculate that our data highlights a role for CIH-induced NOX4-derived ROS in diaphragm muscle dysfunction.

The mechanism underpinning how diaphragm muscle weakness is prevented following exposure to CIH in NOX2 KO mice remains to be elucidated taking the apparent lack of an alteration to NOX2 expression and activity, and the concomitant increase in the abundance of NOX4 into consideration. There is evidence to suggest potential cross-talk between NOX4- and NOX2-dependent ROS in skeletal muscle, however the underlying mechanism and implications of this cascade warrants further exploration [69]. We report an increase in NOX4 mRNA in NOX2 KO mice following exposure to CIH, similar to that observed in wild-type mice following exposure to CIH. We posit that diaphragm muscle weakness does not manifest in NOX2 KO mice following exposure to CIH because the cumulative production of ROS necessary for muscle weakness, possibly due to cross-talk between NOX4 and NOX2, is absent in NOX2 KO mice. Surprisingly, we report no CIH-induced increase in NOX enzyme activity in light of the significant increase in NOX4 expression and we speculate that NOX2-dependent CIH-induced diaphragm dysfunction is based on the acute regulation of NOX2. NAD(P)H consumption assays have been extensively used in the measurement of NOX activity in skeletal muscle studies [22,70,71,72,73]. However, recent evidence suggests that these consumption assays are not specific to NOX activity, as in several tissues and cell types, the signal generated was unaltered when using triple NOX1, NOX2, and NOX4 knock-out [74]. It has been suggested that NOX4 may preferentially utilise NADH as an electron donor (instead of NADPH). Therefore the assay employed in this study may not be optimal to detect alterations in NOX4 activity [75]. Additionally, whole diaphragm homogenates used in the present study may not be sufficient to detect changes in NOX activity as NOX activity has been shown in micro domains of skeletal muscle [69].

Oxidative stress is evident in a variety of tissues across animal models employing CIH, consistent with the fact that human OSAS is an oxidative stress disorder [8]. However, we report no evidence of overt oxidative stress in the diaphragm following exposure to CIH in the present study, despite NOX-dependent ROS underpinning CIH-induced diaphragm muscle weakness in our model. Oxidative stress presents due to an overproduction of ROS, a reduction of antioxidant defense, or a culmination of both. Skeletal muscle possesses a significant antioxidant defense system which enables it to respond efficiently to changes in the redox milieu that occur in both health and disease [76]. A depletion in levels of the endogenous antioxidant GSH (indicated by a reduction in the GSSG/GSH ratio) has been reported following exposure to CIH in rat diaphragm muscle [10]. The administration of NAC throughout the exposure to CIH prevented CIH-induced diaphragm muscle weakness and restored the GSH system to homeostatic levels, highlighting the putative role of redox homeostasis in muscle contractile function [10]. NRF2 is a transcription factor involved in controlling the expression of various antioxidant genes [77]. In the present study, the mRNA expression of NRF2 is significantly increased in the diaphragm following exposure to CIH, with the increase ameliorated in mice administered apocynin and NOX2 KO mice. These findings may indicate that NOX2-derived ROS, produced in response to CIH exposure, disrupts redox balance in the diaphragm muscle sufficient to affect muscle physiology below the threshold for overt cellular oxidative stress. The subsequent increase in the mRNA expression of NRF2 may represent a protective mechanism to up-regulate antioxidant defenses, thereby minimising oxidative injury. Consistent with this, a CIH-induced increase in NRF2 has been observed in both renal and cardiac tissue [78,79]. However, a subsequent increase in SOD1, SOD2, or catalase was not observed in the diaphragm following CIH exposure. The effects of IH on diaphragm muscle function are time-dependent [10] and therefore it is plausible to suggest that accompanying IH-induced alterations to cell signalling pathways are also transient and not entirely understood. Evidence shows that NOX2-derived ROS hold the capacity to activate NRF2 [80]. Moreover, the use of antioxidants to specifically inhibit NOX2, while concomitantly activating NRF2, has shown efficacy in a model of traumatic brain injury [80]. We suggest that the CIH-induced NOX2-dependent increase in the mRNA expression of NRF2 in the present study may represent an early adaptive response, ultimately serving to increase antioxidant status in the diaphragm as a defense against CIH-induced NOX2-derived ROS. We acknowledge that these results are only at the mRNA level and that antioxidant protein responses would be interesting to explore.

Evidence suggests that ROS oxidise lipids in a manner dependent on the severity of OSAS [81]. TBARS are commonly used as a marker of lipid peroxidation, an indirect measurement of oxidative stress. Levels of TBARS exhibit a positive correlation to OSAS severity as measured by increased apnoea index [82]. We found no evidence of increased lipid peroxidation in the diaphragm following exposure to CIH in the present study as levels of TBARS were unaltered. This is consistent with previous studies demonstrating unchanged levels of MDA and 4-HNE in the diaphragm following a 2-week exposure to CIH [10]. We suggest that the relatively mild-to-moderate paradigm of CIH utilised in the present study was not sufficient to cause oxidative stress, in the form of lipid peroxidation.

ROS-dependent changes in mitochondrial function and/or capacity can have detrimental effects on skeletal muscle function [83]. Maladaptation of skeletal muscle in response to hypoxia exposure has been demonstrated as a loss of mitochondrial content yielding a reduced oxidative capacity and less endurant muscle phenotype [84]. Citrate synthase activity is a routinely used marker of mitochondrial integrity. Ten days of exposure to IH has been shown to reduce citrate synthase activity in the diaphragm, suggestive of a shift from mitochondrial respiration towards glycolysis [33]. In contrast, we observed no alteration to citrate synthase activity in the CIH-exposed diaphragm, which likely relates to differences in the paradigm of IH utilised between these studies. Similarly, SDH and GPDH activity, are unaltered in rat diaphragm following exposure to CIH [10,85], indicating that alterations to skeletal muscle metabolism are not likely to underlie the altered contractile function of the diaphragm following exposure to CIH. Mitochondria utilise a specialised form of autophagy, known as mitophagy, to protect against excessive ROS. It is well established that an increase in ROS exacerbates mitochondrial dysfunction, yielding unfavourable consequences for muscle contractile performance. During eccentric exercise ex vivo, inhibiting mitochondrial ROS increases myofibre damage with concomitant muscle weakness [86]. This indicates a potential role for mitochondrial ROS in repair mechanisms in muscle in response to a stressor. In the present study, we examined the mRNA expression of the PINK1/PARK2 pathway, widely considered to be the principal regulator of mitochondrial degradation [87]. Diaphragm PARK2 mRNA expression was increased following exposure to CIH. Both apocynin treatment and NOX2 knockout prevented the increase. Mitophagy as a protective mechanism has been observed in platelets, cardiomyocytes and models of cerebral ischaemia [88,89,90,91]. However, an over-activation of mitophagy in the presence of a persistent stimulus can result in the degradation of essential mitochondrial components, which would similarly have a detrimental effect on skeletal muscle function [92,93]. NOX-derived ROS have been shown to cause an increase mitochondrial ROS, highlighting potential cross-talk between NOX and the mitochondria [94,95]. We postulate that the CIH-induced increase in PARK2 mRNA observed in the present study may be part of a protective mechanism against excessive NOX2-derived ROS-induced mitochondrial stress. However, further studies to assess mitophagic flux and/or dynamics are necessary to confirm the implications of the observed CIH-induced increase in PARK2 mRNA. However, mitochondrial density in CIH-exposed diaphragm is likely normal since citrate synthase activity was unchanged.

Muscle atrophy has not been reported following a 2-week exposure to CIH in the diaphragm in previous studies, however, an increase in type 2B fibres has been observed suggestive of a switch towards a more fatigable muscle phenotype [10,85]. In contrast, four days of IH induced a fibre-type transition towards type-1 fibres, representing a compensatory metabolic switch associated with fatigue resistance [36]. We observed a pro-atrophic (atrogin-1 and MuRF1), and pro-autophagic (LC3B, GABARAPL1 and BNIP3) response at the mRNA level in CIH-exposed diaphragm. These changes were ameliorated by apocynin and were absent in NOX2 KO mice. Similarly, previous studies have demonstrated an increase in the mRNA of autophagy markers (LC3B, GABARAPL1, and BNIP3) with concomitant diaphragm atrophy following four days of IH exposure [36]. Indeed, a loss of muscle mass and subsequent skeletal muscle weakness has been shown to be underpinned by a concomitant increase in autophagy and atrophy [96]. Significant skeletal muscle atrophy with a concomitant increase in NOX4 expression has been observed in rat models of SCI [67]. Similarly, NOX2 deletion prevents Ang-II induced skeletal muscle atrophy [97]. In light of these observations, our results suggest that NOX2-derived ROS could drive diaphragm muscle atrophy, decreasing muscle mass and in this way contribute to reduced force-generating capacity of the diaphragm.

Physiological hypoxia tends to promote myogenesis, whereas pathological hypoxia is largely maladaptive and inhibits myogenesis [98]. The intensity of a hypoxic stimulus appears to be pivotal in determining the myogenic response [99]. NOX2 has been shown to promote muscle differentiation, with this increase prevented by treatment with DPI and apocynin in myoblasts [44]. Consistent with this, we report a largely pro-myogenic response in the diaphragm following exposure to CIH through an increase in the mRNA of key MRFs (Myogenin, MyoD, MEF2C, and Sirtuin-1). Moreover, this increase is absent following co-treatment with apocynin and NOX2 knock out. Persistent rounds of muscle degeneration and regeneration can result in the incomplete remodelling of the ECM. The resultant fibrosis and scar tissue in place of contractile tissue yields a muscle with a lower force-generating capacity [100]. CIH-induced fibrosis has been demonstrated in a multitude of tissues [101,102,103,104,105]. We observed a modest decrease in the shortening velocity and distance of shortening in the diaphragm following exposure to CIH, which is ameliorated following apocynin co-treatment or NOX2 deletion. Therefore, it is plausible to suggest that CIH-induced fibrosis may inhibit optimal contraction. However, we also report a concomitant increase in the negative regulator of myogenesis, myostatin. This is indicative of a decrease in myogenesis, which may serve as a regulated and adaptive response to muscle damage elicited by exposure to CIH. Further work is required to assess diaphragm muscle structure to determine whether transcriptional changes in myogenic signalling translate to structural alterations in the diaphragm muscle and if so, whether changes, if any, are adaptive or maladaptive.

We have confirmed that NOX2-derived ROS underlie CIH-induced diaphragm muscle weakness. We posit that a complex interplay between NOX4 and NOX2 culminates in a sufficient level of ROS necessary to induce diaphragm muscle weakness. Oxidative stress per se is absent in the diaphragm following the paradigm of exposure to CIH utilised in the current study. However, an apparent increase in the antioxidant defense system may represent an early adaptive response to protect against excessive NOX2-derived ROS, perhaps resulting in a modest disruption to redox balance. NOX2-derived ROS appear to drive CIH-induced increases in the expression of genes relating to mitophagy, autophagy, atrophy and myogenesis. Additionally, these processes do not appear to be reliant on HIF-1α. We also posit that NOX-derived ROS may alter diaphragm muscle function through alterations to Ca^2+^ sensitivity of myofilaments thereby affecting cross-bridge cycling or through alterations to Ca^2+^ sensitivity of the contractile apparatus through redox modulation of troponin [106,107,108,109,110,111,112,113]. Alternatively, or in combination, NOX-derived ROS may function through downstream signalling processes involved in the regulation of the structure and thus functional capacity of skeletal muscle.

Antioxidants have shown promise as an adjunct therapy in the treatment of OSAS. OSAS patients treated with vitamin E and C had reduced levels of lipid peroxidation and restored glutathione levels to those reminiscent of healthy controls, demonstrating their efficacy in preventing OSAS-associated oxidative stress [114]. Moreover, this combinational antioxidant treatment reduced apnoea index and excessive day-time sleepiness, the most common symptom of OSAS which significantly impacts OSAS patients’ quality of life [114]. Similarly, OSAS patients treated with NAC exhibited decreased lipid peroxidation and increased levels of glutathione [115]. NAC treatment significantly improved sleep and respiratory parameters indicated by reduced apnoea index, apnoea related-arousals, oxygen desaturations per hour and Epworth sleepiness score [115]. It is suggested that treatment with NAC over a prolonged period of time may reduce patient dependency on CPAP therapy [115]. Similarly, oral carbocysteine treatment improves the Epworth sleepiness score, apnoea index, and percentage of oxygen desaturation, all of which are indicative of a reduction in the severity of OSAS phenotype [116]. Carbocysteine treatment reduced markers of oxidative stress in OSAS patients, evidenced by decreased levels of MDA and increased levels of the endogenous antioxidant, SOD [116]. While the beneficial effects of CPAP treatment and carbocysteine are quite similar, patient compliance was notably higher in OSAS patients treated with carbocysteine compared with CPAP [116]. This highlights that antioxidant treatment is a promising approach for the treatment of OSAS patients who cannot tolerate CPAP.

## 5. Conclusions

We conclude that exposure to CIH, a hallmark feature of human OSAS, is deleterious to diaphragm muscle function, potentially affecting ventilatory and non-ventilatory performance. Combined with upper airway muscle dysfunction [25], exposure to CIH could establish a vicious cycle serving to exacerbate respiratory morbidity in human OSAS. Therapeutic strategies aiming to improve diaphragm muscle performance, via blockade of NOX2, may function as an adjunctive therapy to reduce morbidity in human OSAS. Our results may also have relevance to a variety of other diseases characterised by diaphragm muscle weakness.

## Figures and Tables

**Figure 1 cells-12-01834-f001:**
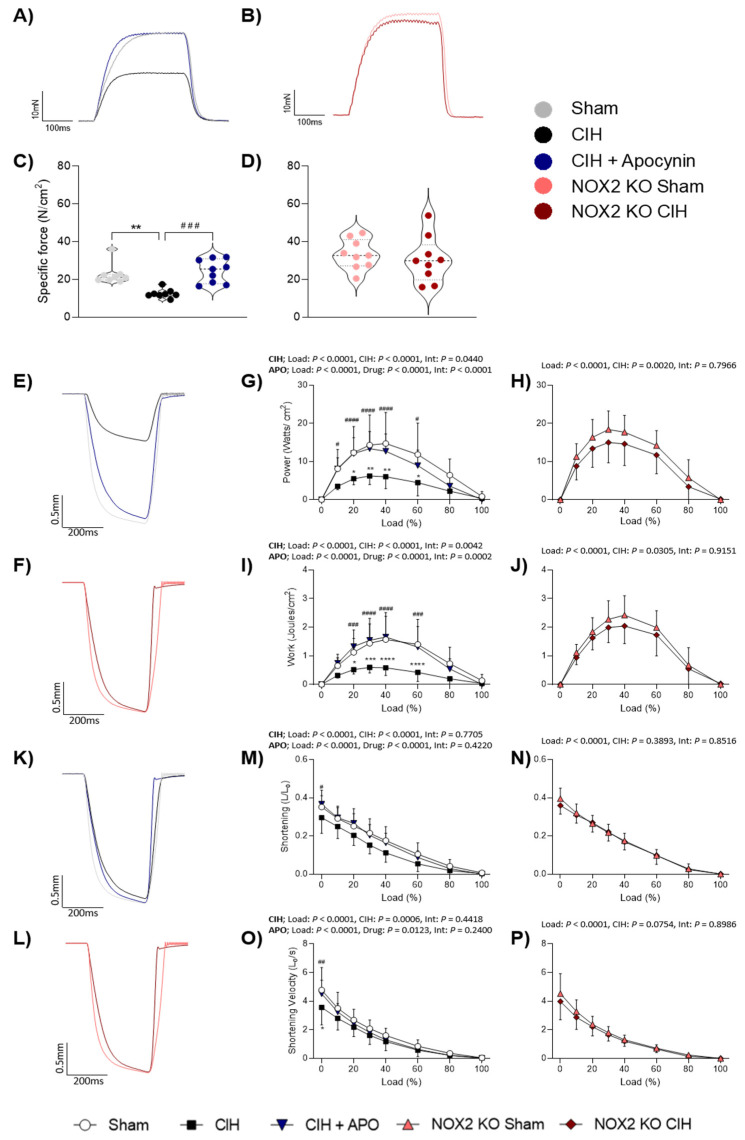
Diaphragm muscle contractile function. **A**) raw traces of diaphragm muscle tetanic cont ractions ex vivo following stimulation at 100 Hz for sham (normoxia (21% O_2_) exposed; grey), CIH (chronic intermittent hypoxia exposed; black), and CIH + APO (apocynin (2 mM) given in the drinking water for the duration of CIH exposure; blue). **B**) raw traces of diaphragm muscle tetanic contractions ex vivo for NOX2 KO sham (NADPH Oxidase 2 knock-out (B6.129S-Cybbtm1Din/J); pink) and NOX2 KO CIH (NADPH Oxidase 2 knock-out (B6.129S-Cybbtm1Din/J); red) experiments. **C**,**D**) group data for normalized diaphragm muscle tetanic force in sham (n = 8), CIH (n = 8), CIH + APO (n = 9), NOX2 KO sham (n = 9) or CIH-exposed (n = 9) mice. Data are shown as violin plots (individual data points, median and interquartile ranges) or as mean ± SD. For **C**), one-way ANOVA with Tukey’s post hoc test was used to statistically compare data sets with confirmed normal distribution. A non-parametric Kruskal–Wallis with Dunn’s post hoc test was used to compare data sets, which were not normally distributed. *p* < 0.05 was deemed statistically significant. For **D**), an unpaired two-tailed Student’s *t* test was used to statistically compare data sets, which were normally distributed. In the case of unequal variance, Welch’s correction was applied. Mann–Whitney non-parametric tests were used to statistically compare data which were not normally distributed. *p* < 0.05 was deemed statistically significant. **E**,**K**), raw traces of diaphragm muscle maximum unloaded shortening ex vivo for sham (grey), CIH (black), and CIH + APO (blue) experiments. **F**,**L**), raw traces of diaphragm muscle maximum unloaded shortening ex vivo for NOX2 KO Sham (pink) and NOX2 KO CIH (red) experiments. For **G**,**H**,**I**,**J**,**M**,**N**,**O**,**P**) repeated measures two-way ANOVA with Bonferroni post hoc test was used to statistically compare data sets. For repeated measures two-way ANOVA; CIH indicates sham vs. CIH; APO indicates CIH vs. CIH + APO for **G**,**I**,**M**,**O**). Int indicates the interaction between two factors for **G**–**P**). *p* < 0.05 was deemed statistically significant for Bonferroni post hoc test. Comparisons are indicated as follows: * indicates sham vs. CIH, * *p* < 0.05, ** *p* < 0.01, *** *p* < 0.001, **** *p* < 0.0001; ^#^ indicates CIH vs. CIH + APO, ^#^
*p* < 0.05, ^##^
*p* < 0.01, ^###^
*p* < 0.001, ^####^
*p* < 0.0001.

**Figure 2 cells-12-01834-f002:**
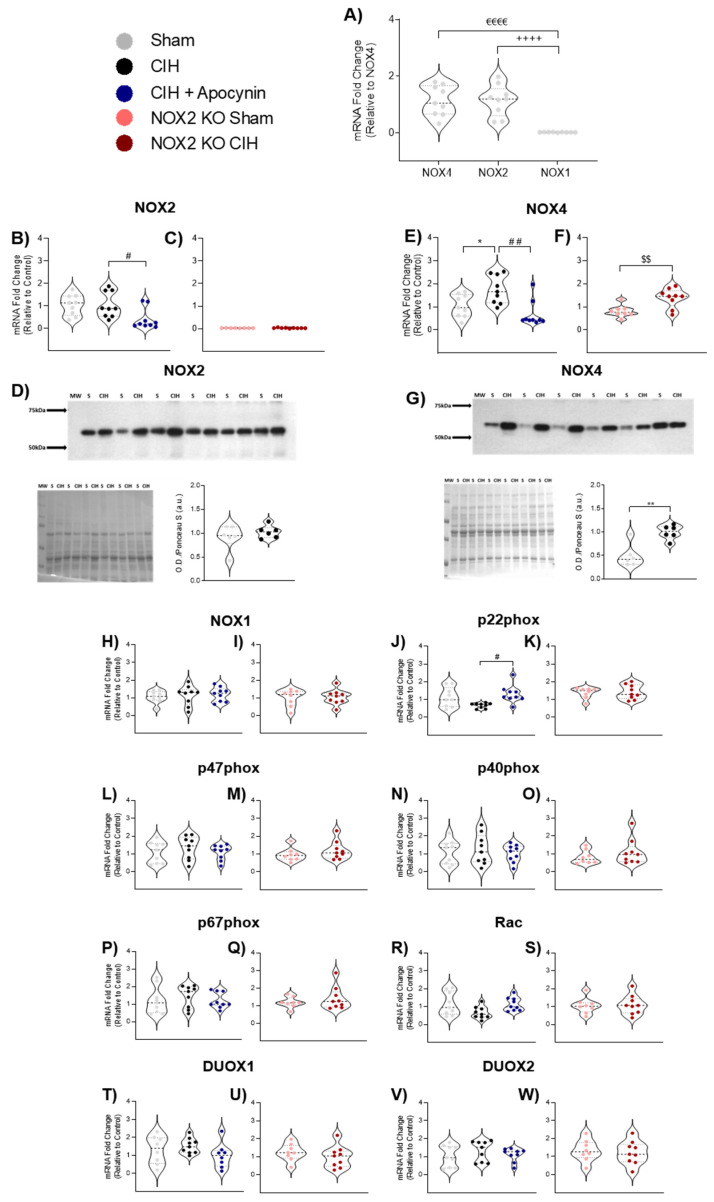
mRNA and protein expression of NOX enzymes in diaphragm muscle. **A**) group data (n = 8–9) shown as a fold change in messenger RNA (mRNA) expression (relative to NOX4) for naïve wild-type mouse diaphragm muscle. **B**,**C**) group data (n = 8–9) shown as a fold change in messenger RNA (mRNA) expression (relative to the control group) for sham, CIH, CIH + APO, NOX2 KO sham, and NOX2 KO CIH groups. **D**) Western Blot of NOX2 protein in diaphragm homogenates from mice exposed to normoxia (sham) or CIH for two weeks, corresponding membrane stained with Ponceau-S and group data (n = 6) for normalised NOX2 expression expressed as arbitrary units (a.u.). **E**,**F**) group data (n = 8–9) shown as a fold change in messenger RNA (mRNA) expression (relative to the control group) for sham, CIH, CIH + APO, NOX2 KO sham, and NOX2 KO CIH groups. **G**) Western Blot of NOX4 protein in diaphragm homogenates from mice exposed to normoxia (sham) or CIH for two weeks, corresponding membrane stained with Ponceau-S and group data (n = 6) for normalised NOX4 expression expressed as arbitrary units (a.u.). **H**–**W**) group data (n = 8–9) shown as a fold change in messenger RNA (mRNA) expression (relative to the control group) for sham, CIH, CIH + APO, NOX2 KO sham, and NOX2 KO CIH groups for **H,I**) NOX1; **J,K**) p22phox; **L,M**) p47phox; **N**,**O**) p40phox; **P**,**Q**) p67phox; **R**,**S**) Rac; **T**,**U**) DUOX1 and **V**,**W**) DUOX2. Data are shown as violin plots (individual data points, median, and interquartile ranges). For **A**,**B**,**E**,**H**,**J**,**L**,**N**,**P**,**R**,**T**,**V**), one-way ANOVA with Tukey’s post hoc test was used to statistically compare data sets with confirmed normal distribution. A non-parametric Kruskal–Wallis with Dunn’s post hoc test was used to statistically compare data sets, which were not normal distributed. All *p* values are adjusted to account for multiple comparisons. *p* < 0.05 was deemed statistically significant. For **C**,**F**,**D**,**G**,**I**,**K**,**M**,**O**,**Q**,**S**,**U**,**W**), an unpaired two-tailed Student’s *t* test was used to statistically compare data sets, which were normally distributed. In the case of unequal variance, Welch’s correction was applied. Mann–Whitney non-parametric tests were used to statistically compare data, which were not normally distributed. *p* < 0.05 was deemed statistically significant. ^€^ indicates NOX4 vs. NOX1, ^€€€€^
*p* < 0.0001; ^+^ indicates NOX2 vs. NOX1, ^++++^
*p* < 0.0001; * indicates sham vs. CIH, * *p* < 0.05. ** *p* < 0.01; ^#^ indicates CIH vs. CIH + APO, ^#^
*p* < 0.05, ^##^
*p* < 0.01; ^$^ indicates NOX2 KO Sham vs. NOX2 KO CIH, ^$$^
*p* < 0.01. Abbreviations: NOX4, NADPH oxidase 4; NOX2, NADPH oxidase 2; NOX1, NADPH oxidase 1; MW, molecular weight marker; sham, normoxia (21% O_2_)-exposed; CIH, chronic intermittent hypoxia-exposed; CIH + APO, Apocynin (2 mM) administered in the drinking water throughout the CIH exposure; NOX2 KO, NADPH Oxidase 2 knock-out (B6.129S-Cybbtm1Din/J); sham, normoxia-exposed (21% O_2_); CIH, chronic intermittent hypoxia-exposed.

**Figure 3 cells-12-01834-f003:**
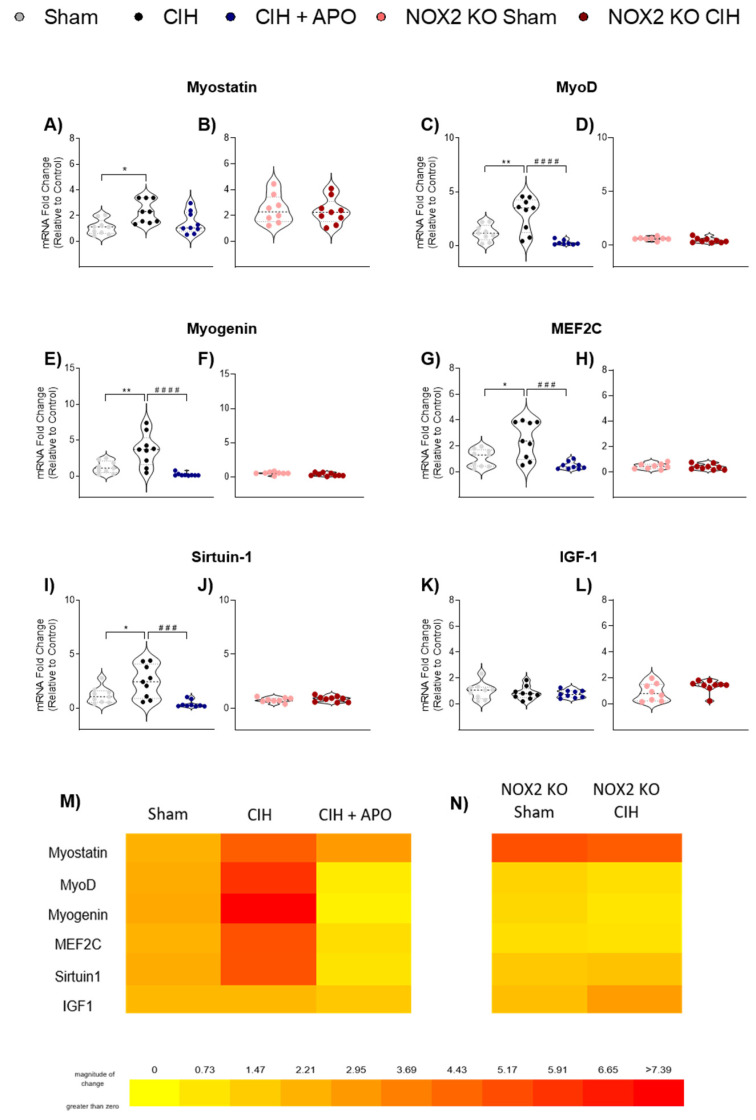
mRNA expression of genes involved in muscle differentiation in diaphragm muscle. Group data (n = 8–9) shown as a fold change in messenger RNA (mRNA) expression (relative to the control group) for sham, CIH, CIH + APO, NOX2 KO sham, and NOX2 KO CIH groups for **A**,**B**) Myostatin; **C**,**D**) MyoD; **E**,**F**) Myogenin; **G**,**H**) MEF2C; **I**,**J**) Sirtuin-1; and **K**,**L**) IGF-1. Heat map illustrating the fold changes in mRNA expression for **M**) Sham, CIH, CIH + APO, **N**) NOX2 KO Sham, and NOX2 KO CIH groups. Warmer colours (red) denote an increase in expression. Data are shown as violin plots (individual data points, median, and interquartile ranges). For **A**,**C**,**E**,G,**I**,**K**), one-way ANOVA with Tukey’s post hoc test was used to statistically compare data sets with confirmed normal distribution. A non-parametric Kruskal–Wallis with Dunn’s post hoc test was used to statistically compare data sets, which were not normally distributed. All *p* values are adjusted to account for multiple comparisons. *p* < 0.05 was deemed statistically significant. For **B**,**D**,**F**,**H**,**J**,**L**), an unpaired two-tailed Student’s *t* test was used to statistically compare data sets, which were normally distributed. In the case of unequal variance, Welch’s correction was applied. Mann–Whitney non-parametric tests were used to statistically compare data, which were not normally distributed. *p* < 0.05 was deemed statistically significant. Relevant comparisons are indicated as follows: * indicated sham vs. CIH; * *p* < 0.05, ** *p* < 0.01; ^#^ indicates CIH vs. CIH + APO; ^###^
*p* < 0.001, ^####^
*p* < 0.0001. Abbreviations: MyoD, muscle differentiation protein 1; MEF2C, myocyte-specific enhancer factor 2C; IGF-1, insulin-like growth factor 1.

**Figure 4 cells-12-01834-f004:**
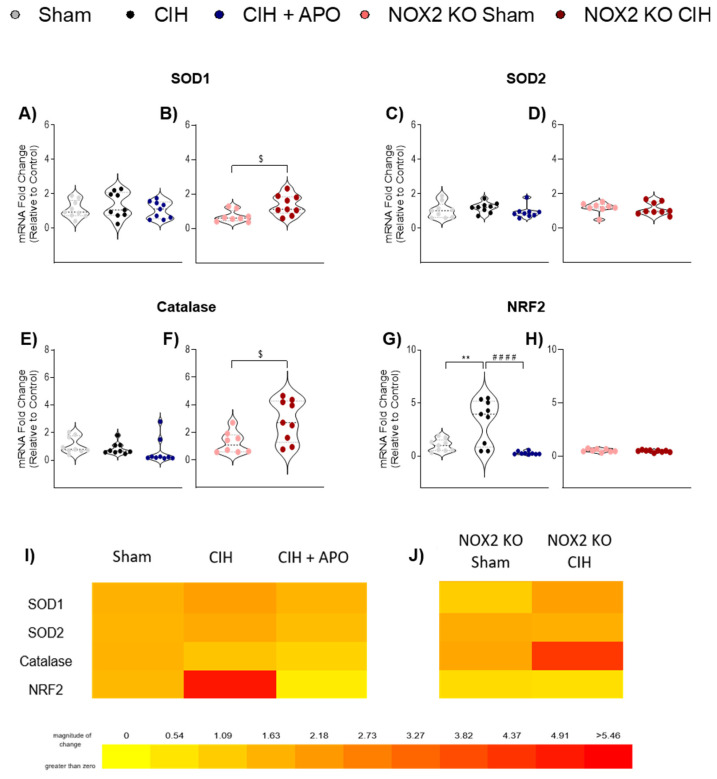
mRNA expression of genes involved in the antioxidant response in diaphragm muscle. Group data (n = 8–9) shown as a fold change in messenger RNA (mRNA) expression (relative to the control group) for sham, CIH, CIH + APO, NOX2 KO sham, and NOX2 KO CIH groups for **A**,**B**) SOD1; **C**,**D**) SOD2; **E**,**F**) Catalase; and **G,H**) NRF2. Data are shown as violin plots (individual data points, median, and interquartile ranges). Heat map depicting the fold changes in mRNA expression for **I**) Sham, CIH, CIH + APO, **J**) NOX2 KO Sham, and NOX2 KO CIH groups. Warmer colours (red) indicate an increase in expression. For **A**,**C**,**E**,**G**), one-way ANOVA with Tukey’s post hoc test was used to statistically compare data sets with confirmed normal distribution. A non-parametric Kruskal–Wallis with Dunn’s post hoc test was used to statistically compare data sets, which were not normally distributed. All *P* values are adjusted to account for multiple comparisons. *p* < 0.05 was deemed statistically significant. For **B**,**D**,**F**,**H**), an unpaired two-tailed Student’s *t* test was used to statistically compare data sets, which were normally distributed. In the case of unequal variance, Welch’s correction was applied. Mann–Whitney non-parametric tests were used to compare data, which were not normally distributed. *p* < 0.05 was deemed statistically significant. * indicates sham vs. CIH; ** *p* < 0.01; ^#^ indicates CIH vs. CIH + APO; ^####^
*p* < 0.0001; ^$^ indicates NOX2 KO Sham vs. NOX2 KO CIH; ^$^
*p* < 0.05. Abbreviations: SOD1, superoxide dismutase 1; SOD2, superoxide dismutase 2; NRF2, nuclear factor erythroid 2-related factor 2.

**Figure 5 cells-12-01834-f005:**
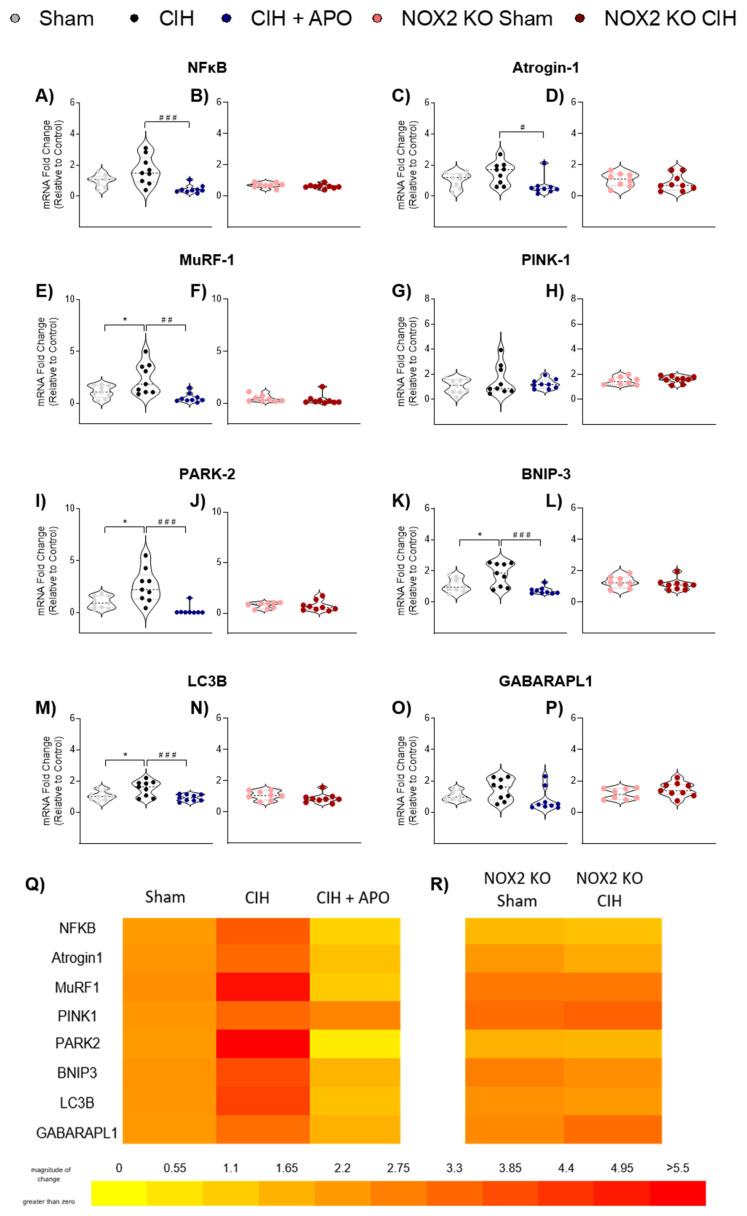
mRNA expression of genes involved in inflammatory and protein degradation processes in diaphragm muscle. Group data (n = 8–9) shown as a fold change in messenger RNA (mRNA) expression (relative to the control group) for sham, CIH, CIH + APO, NOX2 KO sham, and NOX2 KO CIH groups for **A**,**B**) NF-κB, **C**,**D**) Atrogin-1, **E**,**F**) MuRF1, **G**,**H**) PINK1, **I**,**J**) PARK2, **K**,**L**) BNIP3, **M**,**N**) LC3B, **O**,**P**) GAPABARPL1. Data are shown as violin plots (individual data points, median, and interquartile ranges). Heat map depicting the fold changes in mRNA expression for **Q**) Sham, CIH, CIH + APO, **R**) NOX2 KO Sham, and NOX2 KO CIH groups. Warmer colours (red) indicate an increase in expression. For **A**,**C**,**E**,**G**,**I**,**K**,**M**,**O**), one-way ANOVA with Tukey’s post hoc test was used to statistically compare data sets with confirmed normal distribution. A non-parametric Kruskal–Wallis with Dunn’s post hoc test was used to statistically compare data sets, which were not normally distributed. All *P* values are adjusted to account for multiple comparisons. *p* < 0.05 was deemed statistically significant. For **B**,**D**,**F**,**H**,**J**,**L**,**N**,**P**), an unpaired two-tailed Student’s *t* test was used to statistically compare data sets, which were normally distributed. In the case of unequal variance, Welch’s correction was applied. Mann–Whitney non-parametric tests were used to statistically compare data, which were not normally distributed. *p* < 0.05 was deemed statistically significant. Relevant comparisons are indicated as follows: * indicates sham vs. CIH, * *p* < 0.05; ^#^ indicates CIH vs. CIH + APO, ^#^
*p* < 0.05, ^##^
*p* < 0.01, ^###^
*p* < 0.001. Abbreviations: NF-κB, nuclear factor kappa-light-chain-enhancer of activated B cells; MuRF1, muscle RING finger 1; PINK1, PTEN-induced kinase 1; PARK2, parkin; BNIP3, bcl-2 nineteen-kilodalton interacting protein 3; LC3B, microtubule-associated proteins 1A/1B light chain 3B; GABARAPL1, gamma-aminobutyric acid receptor-associated protein-like 1.

**Table 1 cells-12-01834-t001:** Assay details for genes of interest. Real-time ready catalog and custom assays from Roche D used for cDNA amplification.

Gene Name	Gene Symbol	Assay ID
**NOX enzymes**		
*NOX1*	NOX1	310986
*NOX2*	Cybb	317885
*NOX4*	NOX4	300795
*p22phox*	Cyba	317890
*p47phox*	Ncf1	301105
*p67phox*	Ncf2	317897
*p40phox*	Ncf4	317894
*Rac*	Racgap1	310907
*Duox1*	Duox1	317891
*Duox2*	Duox2	317888
**Atrophy**		
*Atrogin-1*	Fbxo32	317844
*MuRF-1*	Trim63	317843
**Autophagy**		
*BNIP3*	Bnip3	311465
*LC3B*	Map1lc3b	317920
*GABARAPL1*	Gabarapl1	317923
**Mitophagy**		
*PINK-1*	Pink1	331846
*PARK-2*	Park2	317264
**Inflammation**		
*NFκB*	Nfkb1	300085
**Antioxidant**		
*SOD1*	Sod1	310738
*SOD2*	Sod2	310295
*Catalase*	Cat	310718
*Nrf2*	Nfe2l2	313377
**Muscle differentiation**		
*Myogenin*	Myog	313501
*Myostatin*	Mstn	318626
*MyoD*	Myod1	313570
*MEF2C*	Mef2c	318629
*IGF1*	Igf1	313359
*Sirtuin-1*	Sirt1	310480
**Reference**		
*HPRT1*	Hprt1	307879

**Table 2 cells-12-01834-t002:** Diaphragm muscle contractile kinetics ex vivo. All values are shown as mean ± SD. One-way ANOVA with Tukey’s post hoc test was used to statistically compare data sets with confirmed normal distribution. A non-parametric Kruskal–Wallis with Dunn’s post hoc test was used to compare data sets, which were not normally distributed. All *p* values are adjusted to account for multiple comparisons. Statistical significance (*p* < 0.05) is indicated by emboldened numbers. Abbreviations: P_t_, twitch force; CT, contraction time; 1⁄2 RT, half-relaxation time; Wmax, maximum work; Pmax, maximum power; Smax, maximum unloaded shortening; Vmax, maximum shortening velocity; L_o_, optimum length; sham, normoxia (21% O_2_)-exposed; CIH, chronic intermittent hypoxia-exposed; CIH + APO, Apocynin (2 mM) administered in the drinking water throughout the 2-week CIH exposure.

	Sham (n = 8)	CIH (n = 8)	CIH + APO (n = 9)	One-Way ANOVA	Sham vs. CIH (*p* Value)	CIH vs. CIH + APO (*p* Value)
Pt (N/cm^2^)	2.62 ± 0.90	1.63 ± 0.31	4.10 ± 1.86	**0.0021**	0.2738	**0.0016**
CT (ms)	15.70 ± 0.96	16.70 ± 3.87	18.17 ± 3.14	0.2378	-	-
½ RT (ms)	19.69 ± 7.82	19.63 ± 6.39	17.50 ± 3.79	0.7045	-	-
Wmax (J/cm^2^)	1.60 ± 0.85	0.63 ± 0.21	1.69 ± 0.84	**0.0012**	**0.0060**	**0.0030**
Pmax (W/cm^2^)	14.70 ± 8.13	6.57 ± 2.35	13.60 ± 4.55	**0.0142**	**0.0193**	**0.0392**
Smax (L/L_0_)	0.35 ± 0.06	0.30 ± 0.08	0.37 ± 0.07	0.1374	-	-
Vmax (L_0_/s)	4.78 ± 1.57	3.59 ± 1.19	4.51 ± 0.95	0.1403	-	-
Muscle mass (mg)	1.48 ± 0.13	1.30 ± 0.21	1.47 ± 0.47	0.2680	-	-
Body mass (g)	23.6 ± 1.5	21.4 ± 0.7	23.5 ± 1.6	**0.0014**	**0.0032**	**0.0047**

**Table 3 cells-12-01834-t003:** Diaphragm muscle contractile kinetics ex vivo. All values are shown as mean ± SD. An unpaired two-tailed Student’s *t* test was used to statistically compare data sets, which were normally distributed. In the case of unequal variance, Welch’s correction was applied. A non-parametric Mann–Whitney test was used to compare data, which were not normally distributed. *p* < 0.05 was deemed statistically significant and is indicated by emboldened numbers. Abbreviations: P_t_, twitch force; CT, contraction time; 1⁄2 RT, half-relaxation time; Wmax, maximum work; Pmax, maximum power; Smax, maximum unloaded shortening; Vmax, maximum shortening velocity; L_o_, optimum length; NOX2 KO, NADPH Oxidase 2 knock-out (B6.129S-Cybbtm1Din/J); sham, normoxia-exposed (21% O_2_); CIH, chronic intermittent hypoxia-exposed.

	NOX2 KO Sham (n = 9)	NOX2 KO CIH (n = 9)	NOX2 KO Sham vs. NOX2 KO CIH (*p* Value)
P_t_ (N/cm^2^)	3.50 ± 2.30	5.52 ± 2.75	0.1107
CT (ms)	12.28 ± 2.00	12.44 ± 1.67	0.8502
½ RT (ms)	13.39 ± 5.41	12.56 ± 5.04	0.8482
Wmax (J/cm^2^)	2.42 ± 0.67	2.16 ± 0.38	0.3923
Pmax (W/cm^2^)	18.74 ± 4.61	15.57 ± 5.02	0.1821
Smax (L/L_0_)	0.40 ± 0.05	0.36 ± 0.04	0.1148
Vmax (L_0_/s)	4.53 ± 1.39	3.98 ± 1.27	0.3958
Muscle mass (mg)	2.01 ± 0.89	1.66 ± 0.39	0.2866
Body mass (g)	28.1 ± 2.5	25.4 ± 0.9	**0.0042**

**Table 4 cells-12-01834-t004:** Various molecular indices in diaphragm homogenates from sham and CIH-exposed mice. All values are shown as mean ± SD. An unpaired two-tailed Student’s *t* test was used to statistically compare data sets, which were normally distributed. In the case of unequal variance, Welch’s correction was applied. Mann–Whitney non-parametric tests were used to statistically compare data, which were not normally distributed. Signal/µg represents the raw signal per µg protein used. *p* < 0.05 was deemed statistically significant and is indicated by bolded numbers. Abbreviations: Sham, normoxia (21% O_2_)-exposed; CIH, chronic intermittent hypoxia-exposed; TBARS, thiobarbituric acid reactive substances; HIF-1α, hypoxia-inducible factor 1-alpha; FOXO3a, forkhead box O3a; ERK 1/2, extracellular-signal-regulated kinase 1/2; JNK, c-Jun N-terminal kinase.

	Sham (n = 8)	CIH (n = 8)	Sham vs. CIH (*p* Value)
NADPH Oxidase Activity (nmol/min/µg)	0.26 ± 0.09	0.29 ± 0.06	0.5565
TBARS (nM/mg)	193.2 ± 93.18	151.6 ± 79.53	0.3536
Citrate Synthase Activity (µmole/mg)	0.68 ± 0.20	0.63 ± 0.22	0.5892
HIF-1α (signal/µg)	2.73 ± 0.48	2.62 ± 0.34	0.5982
Phospho-FOXO-3a (signal/µg)	72.81 ± 19.59	63.27 ± 18.09	0.3284
Phopho-ERK1/2 (signal/µg)	54.49 ± 18.05	41.43 ± 17.61	**0.0379**
Phospho-JNK (signal/µg)	61.15 ± 10.34	48.25 ± 10.82	**0.0287**
Phospho-p38 (signal/µg)	13.83 ± 1.18	11.36 ± 1.10	**0.0007**

## Data Availability

Raw data were generated at the Department of Physiology, University College Cork. The data supporting the findings of this study are available from the authors on request.

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
