# Peer review of "Chronic Intermittent Hypoxia-Induced Diaphragm Muscle Weakness Is NADPH Oxidase-2 Dependent"

_cells, 2023, doi:10.3390/cells12141834_

Round 1
Reviewer 1 Report
The paper explores the function of the diaphragm muscle in mice exposed to intermittent hypoxia, a reliable model of sleep apnea, showing that IH-induced weakness in related to NOX2 activity. Complementary pharmacological and genetic (mice KO for NOX2) approches were used to confirm this. A series of markers were also studied in the muscles at the mRNA or protein expression level, that give further clues on some mechanisms involved in the reported effects. The paper is dense, with many data being reported, but overall presentation remains easy to follow, well organized and adequately discussed. I don't have specific comments or suggestion, and the paper appears acceptable in the present form.
Reviewer 2 Report
See attached file

There are just a few typos and grammatical errors that need to be corrected.
Reviewer 3 Report
Based on pharmacological and transgenic mouse model approaches, Drummond et al. report that NADPH oxidase 2 (NOX2)-derived reactive oxygen species (ROS) contribute to CIH- induced diaphragm muscle dysfunction. NOX2 knockout mice showed none of the striking diaphragm muscle weakness observed in wild-type mice following 2 weeks of exposure to CIH, indicating an obligatory role for ROS produced by NOX2 in diaphragm muscle weakness. A role for CIH-induced NOX4 ROS in diaphragm muscle dysfunction is speculated by the authors. Methodology issues and data gaps regarding the invoked mechanisms are listed below.
General comments:
• The authors report that NOX2 mRNA expression was unaltered in wild-type mice following CIH exposure compared with sham treatment (Figure 2D). The WB figure shows a several-fold increase in NOX2 expression.
• Please include WBs for signaling moieties presented in Table 4.
• In light of the lack of differences in contractile kinetics of the diaphragm in response to CIH, the authors suggest that calcium handling in the diaphragm was unaffected by CIH, and it is unlikely that dysregulated SR functioning explained CIG-induced diaphragm weakness (827-831). Yet, there is a potential crosstalk between NOX4- and NOX2-dependent ROS in the skeletal muscle, whereby NOX4-derived ROS induce RyR Ca2+ leakage, increasing the Ca2+ in the junctional space, and enhancing NOX2-derived ROS. The authors attribute dysregulation of cellular processes involved in reduced muscle performance. This contradiction needs a better explanation.
• The authors do not observe any CIH-induced increase in NOX enzyme activity in light of the significant increase in NOX4 expression and speculate that NOX2-dependent CIH-induced diaphragm dysfunction is based on the acute regulation of NOX2. The ROS levels of fresh frozen skeletal muscle sections can be measured instead of speculating about the regulation and function of NOX2 and NOX4. Why can't ROS levels be measured in diaphragm sections (DHE and MitoPY1 staining) or tissue lysates (HPLC analysis of superoxide and Amplex Red measurement of H2O2)?
• Antioxidant protein responses are more meaningful than mRNA levels in Figure 4. Whether catalase expression and activity correlate or differ, and if not, why?
• ROS levels were not significant enough to induce an antioxidant response but potent enough to induce muscle atrophy and mitophagy gene expression. The induction of muscle atrophy genes contradicts published results. NOX2-derived diaphragm muscle atrophy may better be explained by an increase in NOX2 levels shown in Figure 2.
Other comments
Symbols lack legends in Figs. I G-J and M-P.
Reviewer 4 Report
This is a well writen manuscript of a study inverstigating the role of NOX2 /and potentially NOX4) in IH induced muscle weakness. The excellent style and organisation of the manuscript suggest that it is part of a doctoral thesis as indicated by the authors.
Still, although a thorough study the work remains rather descriptive which leaves the underlying mechanisms how ROS cause muscle weakening and the change in gene expression yet untouched.
I have only a few minor questions/remarks:
1. Is the portocolused for IH really transferable to OSA in patients? The cycling is rather frequent...
2. Do the authors have evidence that carbogen in their contraction studies might influenec force maesurements?
3. It is abit unfortunate that ref 25 contains similar results (obvioulsy fromthis study: Or is there a particular reason why the authors have split their results?
4. Why do the absolute values between table 2 and 3 differ for some parameters?
5. Table 4 The unit "signal/ug" is rather unusual - this should also be explained in the legend or expressed as relative values?
6. Line929 is a falsely formatted reference.
Round 2
Reviewer 2 Report
Thank you for addressing each of the concerns. No further comments or suggestions.
Reviewer 3 Report
Needed new data were not presented.